# CPPO: CONTRASTIVE PERCEPTION FOR VISION LANGUAGE POLICY OPTIMIZATION

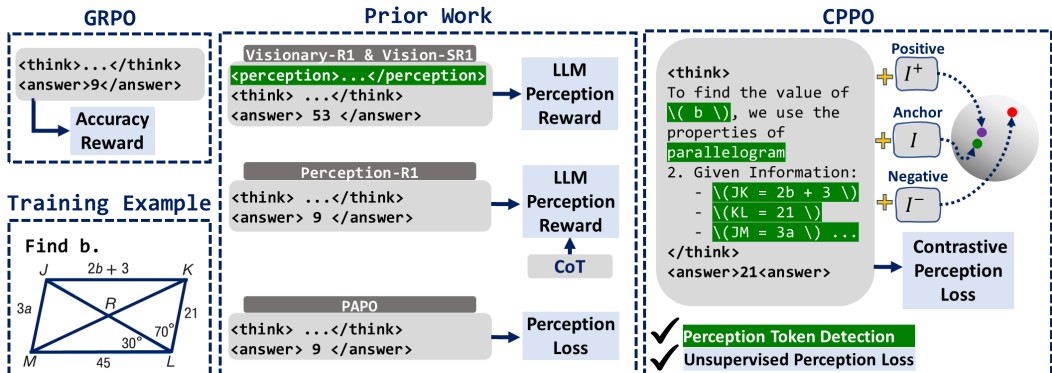

Figure 1: **CPPO vs. prior perception-rewarding methods.** Prior work follows three strategies: (1) Visionary-R1 and Vision-SR1 force the policy to generate separated perception from reasoning, followed by an LLM perception reward, (2) Perception-R1 uses ground-truth CoT and an LLM as a judge to provide perception reward, and (3) PAPO applies a perception loss to all rollout tokens. In contrast, CPPO uses entropy of the output tokens to identify perception tokens and assigns a Contrastive Perception Loss (CPL) exclusively to these tokens.

## ABSTRACT

We introduce **CPPO**, a Contrastive Perception Policy Optimization method for finetuning vision–language models (VLMs). While reinforcement learning (RL) has advanced reasoning in language models, extending it to multimodal reasoning requires improving both the perception and reasoning aspects. Prior works tackle this challenge mainly with explicit perception rewards, but disentangling perception tokens from reasoning tokens is difficult, often requiring extra LLMs, ground-truth data, forced separation of perception from reasoning by policy model, or applying rewards indiscriminately to all output tokens. CPPO addresses this problem by detecting perception tokens via entropy shifts in the model's outputs under perturbed input images. CPPO then extends the RL objective function with a Contrastive Perception Loss (CPL) that enforces consistency under information-preserving perturbations and sensitivity under information-removing ones. Experiments show that CPPO surpasses previous perception-rewarding methods, while avoiding extra models, making training more efficient and scalable. Code is available in the supplementary materials.

## 1 INTRODUCTION

Reinforcement learning (RL) with verifiable rewards has emerged as an effective finetuning method. Notably, DeepSeek-AI (2025) showed the potential of language models to develop reasoning capabilities without explicit step-by-step supervision, focusing on their self-evolution through a pure RL process. Given the success of RL in language models, recent research has focused on extending this approach to vision-language models (VLMs) and multimodal reasoning (Xia et al., 2025; Wang et al., 2025b; Li et al., 2025a; Liu et al., 2025).

In the language-only setting, the policy model draws on its internal knowledge to generate step-by-step logical inference tokens, which we refer to as reasoning tokens. For a VLM policy, however, accurate perception is also required to generate query-relevant factual tokens from the image. We refer to these tokens that encode image information as perception tokens. Wang et al. (2025b) shows that wrong perception tokens are a significant source of failures in multimodal reasoning. However, RL algorithms with verifiable final-answer rewards (e.g., (DeepSeek-AI, 2025)) do not separate perception from reasoning errors. This design is problematic, since inaccurate perception tokens will lead to an incorrect final answer, even with correct reasoning steps. Therefore, achieving the optimal policy is difficult when all output tokens are penalized based on the final answer alone. This limitation raises two questions: *1) How can the output perception and reasoning tokens be disentangled for a VLM policy? 2) How to best define an explicit perception loss/reward?*

To address the first question, Xia et al. (2025) and Li et al. (2025b) force the policy model to separate perception from reasoning, placing perception within `<perception>` tags and reasoning within `<think>` tags. However, forcing a separation between perception and reasoning disrupts the natural reasoning process of the model, making it difficult to apply to many tasks (e.g., with complex images). In addition, the process becomes vulnerable to reward hacking (where the model places the final answer in the perception section to maximize reward). Thus, we argue that perception and reasoning should be disentangled within the natural generation flow of the model.

In order to address the second question, Visionary-R1 Xia et al. (2025), Vision-SR1 (Li et al., 2025b), and Perception-R1 (Xiao et al., 2025) rely on an LLM and utilize either the policy's own perceptual outputs or ground-truth Chain-of-Thought (CoT) annotations to compute perception rewards. Such evaluation of perception outputs with LLMs still require explicit separation of perception from reasoning, incur computational overhead, and rely on unscalable CoT supervision. PAPO (Wang et al., 2025b) takes a different approach via a KL divergence loss between model outputs conditioned on the original and corrupted versions of the images. However, the KL divergence is unbounded, which can easily cause reward collapse and makes the method's hyperparameter sensitive. Moreover, PAPO applies the perception loss uniformly across all tokens and output rollouts, regardless of whether they correspond to perception or reasoning, or whether the outputs are correct or incorrect. Applying divergence over reasoning tokens leads to over-regularization, while maximizing divergence on wrong perception tokens effectively reinforces incorrect perception outputs.

Motivated by these observations, we propose **C**ontrastive **P**erception **P**olicy **O**ptimization (CPPO), an RL solution designed for VLMs. CPPO integrates two main components into the training process: (1) a mechanism that uses policy's own output probability distribution to determine the tokens in a generated response that the policy most strongly considers as perception tokens in its current state, and (2) a token-level Contrastive Perception Loss (CPL) incorporated into the RL objective to enforce *differential sensitivity* to vision information. Specifically, in each training step, we compare the policy's entropy for each token within responses when policy is conditioned on the original image as well as a perturbed image with information-removing augmentations. Tokens whose entropy increases the most under this perturbation are selected as perception tokens by the policy, since their distribution exhibits the highest mutual information with the image.

After identifying vision-dependent tokens in the policy's output, we compute the token-level CPL term. Unlike prior work, CPL is an *unsupervised* perception contrastive loss that does not require additional CoT supervision or proprietary models. Specifically, for each input image, we create two other variants: an information-preserving perturbation that retains query-relevant content and an information-removing perturbation that obscures such information. CPL is then implemented as an InfoNCE contrastive loss (Chen et al., 2020): the token probability distribution conditioned on the original image serves as the anchor, the distribution under the information-preserving perturbation as the positive, and the distribution under the information-removing perturbation as the negative sample. Crucially, the contrastive loss is applied only to perception tokens from *correct* rollouts, ensuring that anchors correspond to accurate and verified perception tokens. This provides targeted perception feedback to the policy, thereby improving its visual grounding capability. In summary, the major contributions of our work are as follows:

- We propose CPPO, an RL-based finetuning solution tailored for VLMs to disentangle perception and reasoning improvement of the policy.
- We propose an entropy-based perception token detection method, where the VLM policy identifies its own perception tokens using its output distribution.

- We propose CPL, an *unsupervised* perception-specific contrastive loss to optimize a VLM policy.
- We show the superiority of CPPO compared with prior perception-specific RL methods.

## 2 RELATED WORK

In this section, we categorize the related RL methods proposed for VLMs into three directions: 1) sampling and rollout augmented methods, 2) RL combined with SFT or off-policy data, and 3) perception-aware approaches. Our approach falls into the third category, while the other directions are orthogonal to our method. We also discuss the background of using contrastive learning in RL.

**Sampling and Rollout Augmented RL with VLMs.** This line of work improves robustness and training efficiency by mixing trajectories from clean and moderately distorted images during RL training. NoisyRollout (Liu et al., 2025) and Vision Matters (Li et al., 2025a) use input perturbations to stabilize grounding and enhance generalization. Shuffle-R1 (Zhu et al., 2025) introduces pairwise trajectory sampling and advantage-based batch reshuffling to improve gradient signal quality and increase exposure to valuable rollouts. VL-Rethinker (Wang et al., 2025a) proposes selective sample replay to address the "vanishing advantages" problem and forced rethinking, which appends a trigger token to enforce self-reflective reasoning. This line of work is orthogonal to CPPO.

**RL Combined with SFT or Off-Policy with VLMs**. This line of research combines on-policy RL with off-policy CoT or SFT training. Vision-R1 (Huang et al., 2025), Look-back (Yang et al., 2025b), OpenVLThinker (Deng et al., 2025), VisionThink (Yang et al., 2025a), and (Shen et al., 2025) focus on semi-off-Policy RL with emphasis on rethinking, iterative pipelines, or off-policy data to enhance slow-thinking reasoning and overall training stability. Similar to the prior category, this line of work is also orthogonal to our work.

**Perception-Aware RL with VLMs.** This category focuses on improving the interaction between perception and reasoning of the policy model. One line of work proposes decoupled architectures such as Guo et al. (2025) and Gou et al. (2025) that use a VLM for visual description and an LLM for reasoning, optimized jointly with RL. Another direction explicitly separates perception from reasoning in the output space of VLMs: Visionary-R1 Xia et al. (2025) and Vision-SR1Li et al. (2025b). Both of these works enforce the policy model to put perception between `<perception>` tokens (or similar tokens) and thinking between `<think>` tokens. The perception tokens are then fed to an LLM to obtain the perception reward. Instead of forcing the model to separate perception from thinking, Perception-R1 Xiao et al. (2025) uses a supervised CoT trajectory to evaluate the perception components of the reasoning trajectory and provides explicit perception rewards. All of these prior works either call an LLM (or the VLM itself) for a second round to answer the question given the perception part or check whether the perception component matched the ground-truth CoT. This design makes inference slower, while also being error-prone when being limited to using small models as a judge. PAPO (Wang et al., 2025b) proposes an additional unsupervised KL divergence loss between the model's outputs conditioned on original and corrupted versions of the images. Such KL divergence has an unbounded nature that can result in collapsed rewards.

**Contrastive Learning in RL.** Contrastive learning has also been explored in RL as a way to learn more robust and discriminative representations. Prior works such as CURL (Laskin et al., 2020), SPR (Schwarzer et al., 2021), SODA (Hansen & Wang, 2021), and TACO (Zheng et al., 2023) leverage contrastive objectives on latent features to improve sample efficiency and generalization in visual RL tasks. Recently, contrastive methods have also been explored for alignment with human feedback. Contrastive Preference Learning (Hejna et al., 2024) proposes learning directly from human feedback signals without relying on standard RLHF pipelines, by using a contrastive objective to distinguish preferred behaviors. Similarly, Contrastive Preference Optimization (CPO) (Xu et al., 2024) applies this principle in the context of LLMs, showing that contrastive objectives can outperform traditional RL-based preference optimization in domains like machine translation. While these methods highlight the versatility of contrastive learning across RL and alignment, VLM policy optimization remains unexplored. Our approach introduces a *token-level contrastive loss* tailored to VLMs, that is applied specifically to vision-dependent tokens within reasoning rollouts.

## 3 METHOD

In this section, we first discuss RL with verifiable rewards in the preliminaries and then elaborate our proposed perception token detection and unsupervised contrastive perception loss.

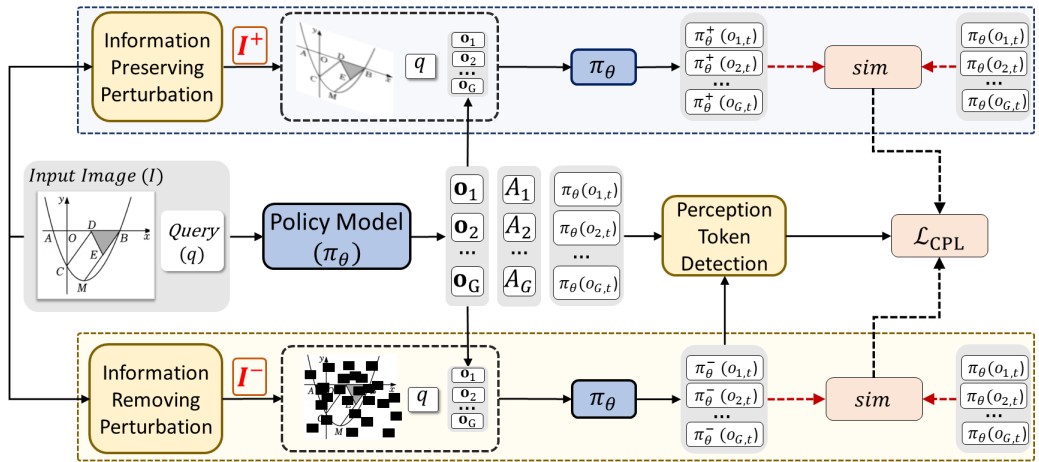

Figure 2: **An overview of CPPO.** For each rollout $\mathbf{o}_i$, perception tokens are identified and their probability distributions are computed under three conditions: the original image $I$ (anchor sample: $\pi_\theta(o_{i,t})$), an information-preserving perturbation $I^+$ (positive sample: $\pi_\theta^+(o_{i,t})$), and an information-removing perturbation $I^-$ (negative sample: $\pi_\theta^-(o_{i,t})$). Similarities $sim\big(\pi_\theta(o_{i,t}), \pi_\theta^+(o_{i,t})\big)$ and $sim\big(\pi_\theta(o_{i,t}), \pi_\theta^-(o_{i,t})\big)$ are computed and incorporated into the Contrastive Perception Loss (CPL), which serves as an additional perception-specific term in the RL objective. Notations are simplified for brevity.

### 3.1 PRELIMINARIES

***Group Relative Policy Optimization* (GRPO).** GRPO (DeepSeek-AI, 2025) includes RL fine-tuning of the policy VLM $\pi_\theta$ with parameters $\theta$ on verifiable tasks. Given an input set $x = \{q, I\}$ including query $q$ and image $I$, a group of $G$ output trajectories (responses) $\{\mathbf{o}_1, \ldots, \mathbf{o}_G\} \sim \pi_\theta(\cdot \mid x)$ are sampled. Each output $\mathbf{o}_i$ consists of $T$ tokens $\{o_{i,1}, \ldots, o_{i,t}, \ldots, o_{i,T}\}$ and receives a scalar reward $R(\mathbf{o}_i)$, typically reflecting correctness. Relative advantages are computed as:

$$A_i = \frac{R(\mathbf{o}_i) - mean\big(R(\mathbf{o}^{1:G})\big)}{std\big(R(\mathbf{o}^{1:G})\big)}, \tag{1}$$

where $i \in [1, G]$. The GRPO objective is then defined as:

$$\mathcal{J}_{\text{GRPO}}(\theta) = \mathbb{E}_{\mathbf{o}_i \sim \pi_{\theta_{old}}} \frac{1}{G} \sum_{i=1}^{G} \frac{1}{|\mathbf{o}_i|} \sum_{t=1}^{|\mathbf{o}_i|}$$

$$\Big\{ min\big( \frac{\pi_\theta(o_{i,t} \mid x, \mathbf{o}_{i,<t})}{\pi_{\theta_{old}}(o_{i,t} \mid x, \mathbf{o}_{i,<t})} A_i, \, clip(\frac{\pi_\theta(o_{i,t} \mid x, \mathbf{o}_{i,<t})}{\pi_{\theta_{old}}(o_{i,t} \mid x, \mathbf{o}_{i,<t})}, 1-\epsilon, 1+\epsilon) A_i \big) - \beta KL[\pi_\theta || \pi_{ref}] \Big\}, \tag{2}$$

where the $KL$ penalty controls the deviation from the frozen reference policy $\pi_{ref}$ with weight $\beta$. Output trajectories are generated by the rollout policy $\pi_{\theta_{old}}$. The hyperparameter $\epsilon$ controls clipping large policy updates. In this setting, the correctness reward alone provides no explicit signal to enhance the policy model's perceptual sensitivity. Our CPL loss aims to address this gap.

### 3.2 CPPO: CONTRASTIVE PERCEPTION POLICY OPTIMIZATION

While prior work has explored guiding the policy model toward improved perceptual understanding by providing explicit vision rewards, our approach is different as it augments the RL objective function with a perception-dependent contrastive loss. Figure 2 illustrates the overall proposed framework. Inspired by contrastive representation learning (Chen et al., 2020), the central idea of CPPO is to encourage the policy to be *differentially sensitive* to visual perturbations in the input image at the *token level*. At each training step, the policy generates a response $\mathbf{o}_i$ for the input $x = \{q, I\}$. Our approach begins by identifying the subset of *perception tokens* within $\mathbf{o}_i$. We then introduce the additional CPL term in RL objective, which is applied to the policy's probability

distribution over the detected perception tokens. Specifically, CPL recomputes policy's probability distribution for each perception token under two variants of the input image $I$:

- **Information-removing perturbations** $I^-$, obtained from transformations such as region masking, occlusion, or deletion of critical visual elements that obscure query-relevant information. The policy's output distribution, $\pi_\theta(o_{i,t} \mid q, I^-, \mathbf{o}_{i,<t})$ should diverge from that of $\pi_\theta(o_{i,t} \mid q, I, \mathbf{o}_{i,<t})$.
- **Information-preserving perturbations** $I^+$, obtained from transformations such as mild Gaussian noise, small brightness shifts, or rotations that do not remove query-relevant content. The output distribution, $\pi_\theta(o_{i,t} \mid q, I^+, \mathbf{o}_{i,<t})$ should remain consistent with that of $\pi_\theta(o_{i,t} \mid q, I, \mathbf{o}_{i,<t})$.

In other words, CPL encourages the model's confidence regarding visual information in a generated response about image $I$ to remain stable if $I$ is altered with irrelevant perturbations, but decrease appropriately when perturbations remove or obscure query-relevant content. Notably, this is achieved in an unsupervised manner, without relying on any CoT data.

### 3.2.1 Perception Token Detection

Not all tokens in an output are equally dependent on perceptual input. For example, interpreting "the base is 10 cm" relies on visual info, whereas solving "$x^2 + 2x + 1 = 0$" or recalling that "the angles of a triangle sum to 180°" can be performed independently of the image. Applying CPL uniformly across all tokens may lead to excessive regularization and destabilize the training. To handle this issue, we propose a mechanism to selectively identify perception-dependent tokens within each trajectory using model's own output distribution. By applying CPL only to these tokens, the model is guided to be sensitive to relevant visual info, while maintaining stability for general reasoning.

**Proposition 1** (**Entropy increase as a proxy for perception dependence**). *(Proof in Appendix A.2) Let $I$ denote the original image, $I^-$ a perturbed variant that removes query-relevant perceptual information, and $\mathbf{o}_i$ the $i$th sequence of tokens generated by the policy when conditioned on $I$. The increase in entropy of a token $o_{i,t} \in \mathbf{o}_i$, when the policy is conditioned on $I^-$ rather than $I$, serves as a proxy for the degree to which the policy model associates $o_{i,t}$ with the query-relevant visual content of $I$. This increase is calculated as follows:*

$$\Delta H_{i,t} = H(o_{i,t}|q, I^-, \mathbf{o}_{i,<t}) - H(o_{i,t}|q, I, \mathbf{o}_{i,<t}). \tag{3}$$

For each token in the $i$th generated sequence of tokens $\{o_{i,1}, \ldots, o_{i,T}\}$, the predictive entropy of the model at position $t$ is defined as:

$$H(o_{i,t}|x, \mathbf{o}_{i,<t}) = -\sum_{o_{i,t} \in \mathcal{V}} \pi_\theta(o_{i,t} \mid x, \mathbf{o}_{i,<t}) \log \pi_\theta(o_{i,t} \mid x, \mathbf{o}_{i,<t}), \tag{4}$$

where $\mathcal{V}$ denotes the vocabulary. This entropy measures the level of uncertainty in predicting the next token based on the input query and image.

**Perception-Top$k$.** We use the criterion in Proposition 1 to identify the most relevant tokens in each response $\mathbf{o}_i$ with respect to the image. After generating $\mathbf{o}_i$ for image $I$, we construct $I^-$ by randomly applying a perturbation from a set of information-removing perturbations. Given $\mathbf{o}_i$ and $I^-$, we compute $\pi_\theta(o_{i,t} \mid q, I^-, \mathbf{o}_{i,<t})$ and measure the corresponding change in entropy ($\Delta H_{i,t}$) for each token $o_{i,t}$. Tokens are then ranked by $\Delta H_{i,t}$, and the top$k$ most perception-dependent tokens are retained. Formally, we define $\mathcal{S}_{\text{perception}}$ as the set of token indices in each response:

$$\mathcal{S}_{\text{perception}} = \{ t \mid \text{Rank}(\Delta H_{i,t}) \leq k \cdot T \}, \tag{5}$$

where $k$ denotes the proportion of tokens with the highest entropy increase, to which that receive the CPL loss is applied. Finally, we construct a binary mask vector $M_i \in \{0,1\}^T$ for the $i$th response:

$$M_{i,t} = \begin{cases} 1, & \text{if } t \in \mathcal{S}_{\text{perception}}, \\ 0, & \text{otherwise.} \end{cases} \tag{6}$$

### 3.2.2 Contrastive Perception Loss (CPL)

After identifying perception tokens via our entropy-based criterion, we now define the token-level CPL. Besides the created $I^-$ with obscured query-relevant information, we generate $I^+$ by sampling

a perturbation from a set of information-preserving perturbations. For each perception token $o_{i,t}$ (i.e., $M_{i,t} = 1$) in each rollout, we treat the policy probability distribution under the original image $I$ as the anchor $\pi_\theta(o_{i,t}) = \pi_\theta(o_{i,t} \mid q, I, \mathbf{o}_{i,<t})$, the distribution under $I^+$ as the positive sample $\pi_\theta^+(o_{i,t}) = \pi_\theta(o_{i,t} \mid q, I^+, \mathbf{o}_{i,<t})$, and the distributions under $I^-$ as the negative sample $\pi_\theta^-(o_{i,t}) = \pi_\theta(o_{i,t} \mid q, I^-, \mathbf{o}_{i,<t})$. Let $sim(p, p^*) = -KL(p \| p^*)$ denote the negative KL divergence as an estimate similarity between token probability distributions. Then, our contrastive loss term is defined by adopting the InfoNCE loss (Chen et al., 2020):

$$\mathcal{L}_{o_{i,t}}^{\text{InfoNCE}} = -\log \frac{\exp\left\{sim\left(\pi_\theta(o_{i,t}), \pi_\theta^+(o_{i,t})\right)/\tau\right\}}{\exp\left\{sim\left(\pi_\theta(o_{i,t}), \pi_\theta^+(o_{i,t})\right)/\tau\right\} + \exp\left\{sim\left(\pi_\theta(o_{i,t}), \pi_\theta^-(o_{i,t})\right)/\tau\right\}}, \quad (7)$$

where $\tau > 0$ is a temperature hyperparameter. Minimizing this loss encourages the anchor distribution $\pi_\theta(o_{i,t})$ to remain close to the positive view $\pi_\theta^+(o_{i,t})$ while being pushed away from the negative view $\pi_\theta^-(o_{i,t})$. For all tokens in the $i$th trajectory $\mathbf{o}_i$, the CPL is defined as:

$$\mathcal{L}_{\text{CPL},i,t} = \begin{cases} \mathcal{L}_{o_{i,t}}^{\text{InfoNCE}} & \text{if } M_{i,t} = 1, \\ 0, & \text{if } M_{i,t} = 0. \end{cases} \quad (8)$$

That is, non-perception tokens ($M_{i,t} = 0$) are excluded from the CPL. The overall CPL for the $i$th trajectory $\mathbf{o}_i$ is obtained by averaging over its tokens:

$$\mathcal{L}_{\text{CPL}}(\mathbf{o}_i; I, I^+, I^-) = \frac{1}{|\mathbf{o}_i|} \sum_{t=1}^{|\mathbf{o}_i|} \mathcal{L}_{\text{CPL},i,t}. \quad (9)$$

**Integration with GRPO.** Finally, we integrate CPL with the GRPO objective. For each sampled trajectory $\mathbf{o}_i$, we compute the standard GRPO update (Eq. 2) along with the CPL. To prevent low-quality trajectories from introducing noisy gradients, we use an advantage gating mechanism, whereby CPL is only applied when the trajectory's group-relative advantage $A_i$ is positive. Formally, we maximize the following combined objective:

$$\mathcal{J}(\theta) = \mathbb{E}_{\mathbf{o}_i \sim \pi_{\theta_{old}}} \left[ \mathcal{J}_{\text{GRPO}}(\theta) - \lambda \frac{1}{G} \sum_{i=1}^{G} \left\{ \mathbf{1}\{A_i > 0\} \cdot \mathcal{L}_{\text{CPL}}(\mathbf{o}_i; I, I^+, I^-) \right\} \right], \quad (10)$$

where $\mathbf{1}\{\cdot\}$ is the indicator function. The hyperparameter, $\lambda$, controls the strength of perceptual grounding. By incorporating the advantage gating mechanism, we ensure that CPL acts as an auxiliary constraint only on trajectories that improve upon the group baseline, thereby aligning visual regularization with successful reasoning behaviors.

# 4 EXPERIMENTS

## 4.1 EXPERIMENTAL SETUP

**Training Dataset.** We train on ViRL39K (Wang et al., 2025a), a dataset consisting of 38.8K multimodal question–answer pairs. The dataset spans a broad range of domains, including grade school problems to broader STEM and social topics; reasoning with charts, diagrams, tables, documents, and spatial relationships.

**Evaluation.** Following prior works, we use the following benchmarks for evaluation: LogicVista (Xiao et al., 2024), MathVista (Lu et al., 2024), DynaMath (Zou et al., 2025), WeMath (Qiao et al., 2024), MathVision (Wang et al., 2024), MathVerse (Renrui Zhang, 2024), and MMMU-Pro-Vision (Yue et al., 2025). These benchmarks encompass math, general multimodal reasoning, and logical reasoning tasks. All evaluations are performed using VLMEvalKit (Duan et al., 2024). We report average accuracy@8 with an inference temperature of 1.0 to provide a more consistent and reliable measure of model performance across all the experiments in the paper.

**Baselines.** We use Qwen2.5-VL-3B and 7B (Bai et al., 2025) as the backbone models in all our experiments. We compare our CPPO with recent RL methods proposed for VLMs: OpenVLThinker-3B/7B, Visionary-R1-3B, PAPO-3B/7B, VL-ReThinker-7B, Vision-Matters-7B, Perception-R1-7B, Vision-SR1-7B, NoisyRollout-7B, and Look-Back-7B (semantic checkpoint). All of these prior works use Qwen2.5-VL-3B/7B as the policy model and, therefore, comparisons are fair.

Table 1: CPPO vs. prior works. All results are based on avg@8. For prior methods, we used their released checkpoints. **Bold**: the best value in each column. Underlined: the second best.

| Methods | Math Benchmarks | | | | | Visual Reasoning | | |
|---|---|---|---|---|---|---|---|---|
| | MVista$_m$ | DMath | WeMath | MVision$_m$ | MVerse | MMMU-P$_v$ | LogicVista | Avg. |
| GPT4-o | 60.0 | 34.5 | 47.4 | 30.6 | 41.2 | 51.9 | 52.8 | 45.4 |
| Gemini-2.0-Flash | 73.4 | 42.1 | 45.8 | 41.3 | 54.6 | 51.7 | 52.3 | 51.6 |
| *Qwen2.5-VL-3B* | 56.4 | 33.7 | 14.5 | 19.5 | 25.7 | 19.9 | 32.4 | 28.8 |
| OpenVLThinker | 60.0 | 35.6 | 26.3 | 22.3 | 36.9 | 25.0 | 37.4 | 34.7 |
| Visionary-R1 | 61.4 | 41.2 | 27.1 | 19.7 | 34.5 | 27.9 | 37.1 | 35.5 |
| PAPO | 64.8 | 45.4 | 28.1 | 24.3 | 38.3 | 26.8 | 39.4 | 38.1 |
| GRPO | 63.7 | 45.7 | 28.4 | 25.1 | 38.3 | 25.8 | 37.7 | 37.8 |
| **CPPO (ours)** | **66.3** | **48.9** | **30.8** | **25.3** | **39.4** | **28.5** | **40.9** | **40.0** |
| *Qwen2.5-VL-7B* | 65.6 | 53.2 | 33.3 | 24.5 | 41.2 | 33.7 | 45.1 | 42.3 |
| OpenVLThinker | 70.7 | 43.9 | 38.4 | 27.5 | 40.7 | 35.5 | 45.8 | 43.9 |
| Vision-SR1 | 67.0 | 52.6 | 33.6 | 28.0 | 40.7 | 38.9 | 43.2 | 43.9 |
| Look-Back | 69.1 | 52.5 | 39.8 | 25.8 | 41.9 | 34.5 | 46.3 | 44.8 |
| Vision-Matters | 68.6 | 54.5 | 40.1 | 25.2 | 45.3 | 35.5 | 45.1 | 45.3 |
| PAPO | 71.6 | 54.7 | 39.5 | 26.5 | 44.5 | 38.7 | 45.8 | 46.8 |
| PerceptionR1 | 70.0 | 55.8 | **45.4** | 27.6 | 46.0 | 38.1 | 45.5 | 47.3 |
| NoisyRollout | 71.1 | 55.9 | 44.4 | 29.4 | 46.4 | 38.5 | 47.9 | 47.7 |
| GRPO | 71.2 | 55.6 | 42.4 | 27.6 | 45.0 | 37.9 | 47.4 | 46.7 |
| **CPPO (ours)** | **72.2** | **56.9** | 44.8 | **29.9** | **46.5** | **39.0** | **48.2** | **48.2** |

**Implementation Details.** We use *verl* (Sheng et al., 2024) as our RL training framework. The policy models are initialized with Qwen2.5-VL-3B/7B. We train the policy model with GRPO and CPPO for 2 epochs on the ViRL39K dataset with a group size of 5 and a global batch size of 512. Both the vision encoder and LLM of the baselines were updated during training. For other RL-related hyperparameters, we use the default settings of *verl*. More details are in the Appendix A.3.

**Perturbation Types.** For information-removing perturbations, we employ random 80% patch-wise masking and random 30% cropping (retaining only 30% of the image) to obscure the majority of the visual content. For information-preserving perturbations, we apply lightweight transformations such as color jitter, random perspective, random rotation, and Gaussian noise, which modify the image appearance without eliminating critical information. At each training step, one augmentation is randomly sampled from each augmentation set. Detailed parameter settings and illustrative examples of all perturbations are provided in the Appendix A.4.

## 4.2 MAIN RESULTS

The performance of closed-source models (GPT4-o, Gemini-2.0-Flash), the backbone models, and RL-based baseline VLMs including GRPO compared with our CPPO is reported in Table 1.

**Comparison to Baseline GRPO.** Applying CPPO to the Qwen2.5-VL-3B and -7B baselines yields consistent and substantial improvements on the test benchmarks, with average performance gains of 11.2% and 5.9%, respectively. As reported in Table 1, CPPO achieves a higher accuracy than GRPO across all benchmarks—average 40.0% vs. 37.8% for the 3B model and 48.2% vs. 46.7% for the 7B model. Overall, these results confirm that CPPO is a more effective optimization strategy than GRPO, especially for mid-sized models, and establishes CPPO as a strong and scalable alternative for finetuning large VLMs. Qualitative results are given in the Appendix A.7.

**Comparison to Other Methods.** As shown in Table 1, CPPO consistently surpasses prior methods across all benchmarks for the 3B model. For the 7B model, CPPO also outperforms existing approaches on all benchmarks (except WeMath), demonstrating stronger generalization. In particular, when compared to PAPO—the most relevant perception-aware RL baseline—CPPO achieves notable gains. On the 3B model, CPPO improves average performance to 40.0%, compared to PAPO's 38.1%. On the larger 7B model, CPPO reaches 48.2% versus PAPO's 46.8%. Importantly,

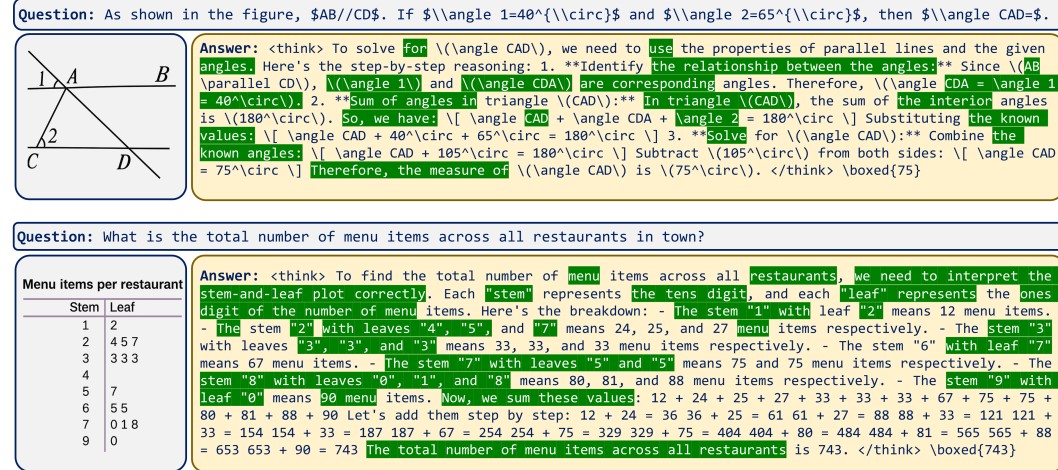

Figure 3: Sample outputs generated with CPPO with top 40% detected perception tokens.

both CPPO and PAPO are trained under identical conditions—using the same dataset (ViRL39K) and the same number of training steps—ensuring that the improvements are not due to differences in data or compute. Thus, the consistent advantage of CPPO over PAPO can be attributed directly to the introduction of contrastive loss on perception tokens, which enhances the model's ability to capture and leverage visual information more effectively.

**Performance of Perception Token Detection.** Figure 3 shows two samples, the policy model's outputs generated by CPPO, and the top 40% of perception tokens identified using our entropy-based method. In the 1st example, the question asks for the value of angle $\angle CAD$ in a geometry problem. The key visual clues needed to solve this question are: (1) $\angle CDA = \angle 1 = 40°$, (2) CAD forms a triangle, and (3) $\angle 2 = \angle ACD$. With these three pieces of information alone, one could solve the problem without referring back to the original figure. We observe that all these critical elements are successfully highlighted within the top 40% of selected perception tokens. The 2nd example shows a stem and leaf plot summarizing the number of menu items per restaurant in a town, which is used to answer a question. Here, we find that most of the relevant numerical values are also captured within the top 40% of detected perception tokens, illustrating that the method effectively identifies the essential visual information for the question. Numerical analysis is given in the Appendix A.5.

## 4.3 ABLATIONS

We adopt Qwen2.5-VL-3B as the baseline model and conduct all ablations on the Geometry3K dataset Lu et al. (2021), which contains 2.1K samples. We select Geometry3K both to enable faster training and to demonstrate the generalizability of our approach across different training datasets.

**Ablation on Main Components of CPPO.** Table 2 reports the ablation study on the key components of CPPO. Starting from GRPO, applying CPL to all tokens raises the average accuracy from 34.7% to 35.0%. Restricting CPL to only the top 50% of perception tokens yields a larger gain, increasing accuracy to 36.6%. Finally, introducing advantage gating—where the contrastive loss is applied only to rollouts with positive advantage—further improves performance to 38.6%. These results highlight that each component makes a meaningful contribution, and together they account for the overall effectiveness of CPPO.

Table 2: Ablation on the main component of CPPO.

| Methods | LogicVista | MVista$_m$ | MVision$_m$ | WeMath | **Avg.** |
|---|---|---|---|---|---|
| *Qwen2.5-VL-3B* | 32.4 | 56.4 | 19.5 | 14.5 | 30.7 |
| GRPO | 35.4 | 55.9 | 20.9 | 26.7 | 34.7 |
| + Contrastive Loss on All Tokens | 35.6 | 56.0 | 20.8 | 27.2 | 35.0 |
| + Contrastive Loss on Top$k$ Perception Tokens | 36.4 | 56.6 | 22.5 | 30.9 | 36.6 |
| + Advantage Gating | 38.5 | 59.9 | 23.1 | 32.9 | **38.6** |

**Top$k$.**    Table 3 presents the analysis of different $K$ values for top$k$ perception token detection. The results show that average accuracy improves as $K$ increases from 5% to 50%, but declines when $K$ is further expanded from 50% to 100%. We hypothesize that this trend arises because larger $K$ values include more tokens that the policy model is already confident about

Table 3: Experiments on top$k$ perception tokens.

| K | LogicVista | MVista$_m$ | MVision$_m$ | WeMath | Avg. |
|---|---|---|---|---|---|
| 5% | 32.5 | 52.2 | 21.4 | 20.1 | 31.6 |
| 25% | 36.7 | 57.9 | 22.7 | 30.7 | 37.0 |
| 50% | 38.5 | 59.9 | 23.1 | 32.9 | **38.6** |
| 75% | 37.6 | 57.4 | 22.3 | 29.1 | 36.6 |
| 100% | 36.3 | 56.9 | 22 | 29.5 | 36.2 |

(i.e., tokens with lower entropy change), which are less informative perception tokens. Incorporating these tokens can slow down the training and ultimately lead to worse performance when models are trained for the same number of epochs.

**Loss Weighting ($\lambda$).**    We experiment with different $\lambda$ values in Eq. 10. $\lambda$ controls the strength of perceptual grounding. As given in Table 4, the best performance is obtained with $\lambda = 0.02$. In general, CPPO with different $\lambda$ values outperforms GRPO with an average accuracy of 34.7% as reported in Table 2.

Table 4: Experiments on $\lambda$ values.

| $\lambda$ | LogicVista | MVista$_m$ | MVision$_m$ | WeMath | Avg. |
|---|---|---|---|---|---|
| 0.01 | 37.4 | 59.2 | 21.9 | 31.4 | 37.5 |
| 0.02 | 38.5 | 59.9 | 23.1 | 32.9 | **38.6** |
| 0.03 | 38.6 | 57.8 | 22.9 | 28.8 | 37.0 |
| 0.04 | 35.6 | 55.9 | 21.7 | 27.6 | 35.2 |

## 4.4    Reward Graph

Figure 4 shows the training dynamics of CPPO vs. GRPO (Training Reward), reward on the in-domain validation set (Geometry3K Validation Reward), and detailed accuracy comparison across out-of-domain benchmarks as training progress. The training reward shows that CPPO leads to faster learning as well as strong out-of-distribution generalization from the early steps of training.

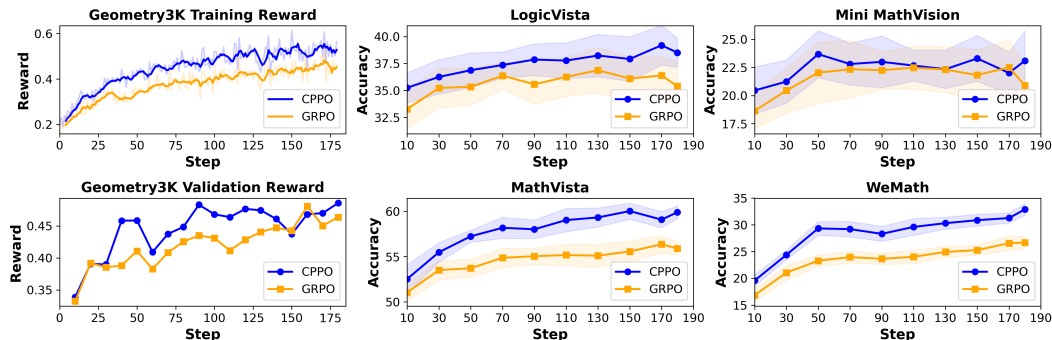

Figure 4: CPPO vs. GRPO (avg@8) on Qwen2.5-VL-3B across in-domain and out-of-domain scenarios. The X-axis represents RL training steps. The shaded area corresponds to one standard deviation. **1st column:** Reward comparison on the in-domain dataset during training. **2nd and 3rd columns:** Comparison on four out-of-domain visual reasoning benchmarks.

## 5    Conclusion

In this work, we introduced CPPO, a perception-aware RL-based method for finetuning VLMs. CPPO leverages an entropy-based approach to disentangle *perception* tokens from *reasoning* tokens, where perception tokens capture visual information extracted from the input image. To better align training with perception quality, we proposed a Contrastive Perception Loss (CPL)—an unsupervised, model-free objective that penalizes perception errors. Extensive experiments demonstrate that CPPO outperforms recent RL methods for VLMs, achieving state-of-the-art performance across multiple math and visual reasoning benchmarks. We discuss the limitations of our approach and directions for future work in the Appendix A.6.

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

# A APPENDIX

In this appendix, we present the following additional discussions and experimental results:

- Code
- Proof for Proposition 1
- Extra Training Details
- Sample of Perturbations
- Further Analysis: Perception Token Detection
- Limitations of Our Work
- Qualitative Results
- Statement on LLMs Assistance

## A.1 CODE

In order for our results to be reproducible, we share our code as supplementary materials, with detailed instructions included in the associated README.md file.

## A.2 PROOF FOR PROPOSITION 1

**Proposition 1 (Entropy increase as a proxy for vision dependence).** *Let $I$ denote the original image, $I^-$ a perturbed variant that removes query-relevant perceptual information, and $\mathbf{o}_i$ the sequence of tokens generated by the policy when conditioned on $I$. The increase in entropy of a token $o_{i,t} \in \mathbf{o}_i$, when the policy is conditioned on $I^-$ rather than $I$, serves as a proxy for the degree to which the policy model associates $o_{i,t}$ with the query-relevant visual content of $I$. This increase is calculated as follows:*

$$\Delta H_{i,t} = H(o_{i,t}|q, I^-, \mathbf{o}_{i,<t}) - H(o_{i,t}|q, I, \mathbf{o}_{i,<t}).$$

*Proof.* Recall the identity relating conditional mutual information (denoted by $MI$) and conditional entropy:

$$H(o_{i,t} \mid X, q, \mathbf{o}_{i,<t}) = H(o_{i,t} \mid q, \mathbf{o}_{i,<t}) - MI(o_{i,t}; X \mid q, \mathbf{o}_{i,<t})). \tag{11}$$

Applying this with both $X = I$ and $X = I^-$ and subtracting, we obtain

$$
\begin{aligned}
H_{i,t}(I^-) - H_{i,t}(I) &= H\big(o_{i,t} \mid I^-, q, \mathbf{o}_{i,<t}\big) - H(o_{i,t} \mid I, q, \mathbf{o}_{i,<t}) \\
&= MI(o_{i,t}; I \mid q, \mathbf{o}_{i,<t}) - MI\big(o_{i,t}; I^- \mid q, \mathbf{o}_{i,<t}\big).
\end{aligned} \tag{12}
$$

$I^-$ is obtained from $I$ by an information-removing augmentation that obscures query-relevant visual information. Our main assumption is that the conditional mutual information between perception tokens in $\mathbf{o}_i$ and $I$ should be greater than their conditional mutual information with the perturbed image $I^-$. Formally, if $o_{i,t}$ is a perception token, we assume the following inequality holds for its conditional mutual information:

$$MI(o_{i,t}; I \mid q, \mathbf{o}_{i,<t}) - MI\big(o_{i,t}; I^- \mid q, \mathbf{o}_{i,<t}\big) \geq 0. \tag{13}$$

Substituting this inequality into equation 12 yields

$$H_{i,t}(I^-) - H_{i,t}(I) \geq 0. \tag{14}$$

Thus, an increase in predictive entropy, $\Delta H_{i,t}$, serves as a principled proxy for identifying vision-dependent tokens in the output sequence. $\square$

## A.3 EXTRA TRAINING DETAILS

Table 5 shows the summary of hyper-parameters used in training of 3B and 7B models.

Table 5: Summary of hyperparameter configurations.

| Parameter | Configuration |
|---|---|
| **Main Results** | |
| Model Base | Qwen2.5-VL-Instruct |
| Global Batch Size | 512 |
| Rollout Temperature | 1.0 |
| Learning Rate | $1e^{-6}$ |
| Rollout Number | 5 |
| Training Epochs | 2 |
| Optimizer | AdamW |
| Policy Loss Aggregation | `token-mean` |
| $\beta$ | 0.01 |
| $\tau$ | 0.1 |
| $k$ | 50% |
| $\lambda$ | 0.02 |
| **Ablations Specific** | |
| Dataset | Geometry3K |
| Training Epochs | 12 |
| Global Batch Size | 128 |

### A.4 SAMPLE OF PERTURBATIONS

Figure 5 illustrates examples from the training dataset along with two categories of perturbations: information-removing and information-preserving. The information-removing perturbations, such as random occlusion and random zoom crop, eliminate key visual details necessary for understanding the image. In contrast, the information-preserving perturbations—including color jitter, random perspective, random rotation, and Gaussian blur—modify the image without discarding critical information.

### A.5 FURTHER ANALYSIS: PERCEPTION TOKEN DETECTION

To quantitatively evaluate our perception detection method, we used the inference outputs of Qwen2.5-VL-3B and -7B on four test sets: MathVista-MINI, LogicVista, MathVision-MINI, and WeMath. We then passed these outputs to GPT5-mini, which was used to separate the perception-related information from the rest of the model's response. This extracted perception information serves as our ground truth. We measure the accuracy of our detection method by calculating the ROUGE-1 F1 score between the detected perception to-

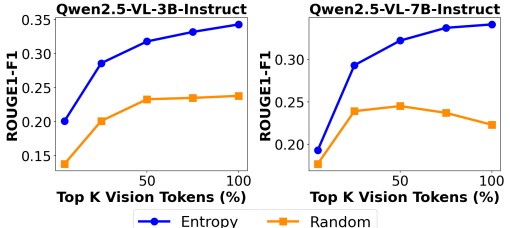

Figure 6: Quantitative evaluation of perception token detection.

kens and the GPT5-mini outputs. It is important to note that GPT5-mini's separation is not flawless; thus, this evaluation should be viewed as a proof-of-concept rather than a definitive benchmark. Figure 6 shows that the ROUGE-1 F1 score improves as we increase the number of top-$k$ perception tokens, up to the point where 100% of perception tokens are included. Here, 100% refers to selecting all tokens with positive $\Delta H$ in Proposition 1, rather than all output tokens. At each top-$k$ percentage, we also select the same number of tokens randomly to serve as a baseline. Figure 6 shows that there is significant gap between our entropy-based method and random selection.

### A.6 LIMITATIONS

This work has several limitations that should be addressed in future research. First, due to our computational constraints, we did experiments up to 3B and 7B models. Exploring larger VLMs, such as 72B models, is an important direction for future work. Second, our evaluation was limited to

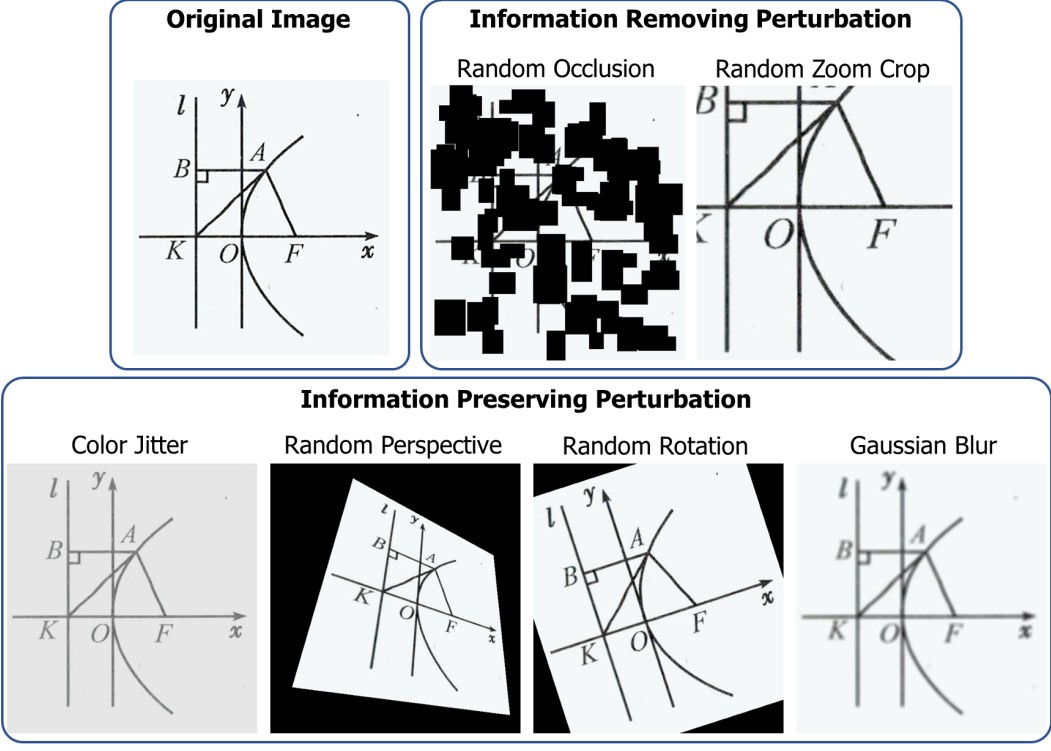

Figure 5: Sample information-removing perturbations and information-preserving perturbations.

Qwen2.5-VL baselines; extending the analysis to other baselines, such as InternVL (Chen et al., 2024), would provide a more comprehensive comparison. Finally, while we demonstrated the effectiveness of CPPO using 40K training samples, future studies should investigate large-scale training with substantially larger datasets.

### A.7 QUALITATIVE RESULTS

Figures 7-9 show three qualitative examples. We observe that CPPO has corrected the perception mistakes of models trained with GRPO. For example, in Figure 7, the model trained with GRPO states that `"the angle x is given as 70 degrees"` that is a wrong perception information extracted from the image. However, the model trained with CPPO corrected this statement by `"The two line segments form angles that add up to 180 degrees"`. Note that when perception tokens are wrong, even with correct reasoning trajectory, the final answer is wrong.

### A.8 STATEMENT ON LLMS ASSISTANCE

We declare that some portions of this document have been lightly refined using Large Language Models (e.g., ChatGPT) to enhance clarity and polish. All substantive content and ideas remain entirely our own.

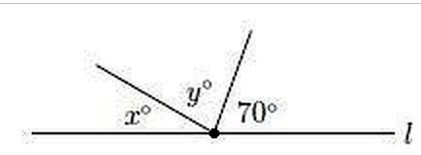

Note: Figure not drawn to scale.

Hint: Please answer the question requiring an integer answer and provide the final value, e.g., 1, 2, 3, at the end.
Question: In the figure above, two line segments meet at a point on line l. If the value of y is equal to the square of the value of x, what is the value of y?

**CPPO**

<think>
The two line segments form angles that add up to 180 degrees because they are supplementary angles. This means that the sum of their measures is 180 degrees. Therefore, we can write the following equation:
$$ x^\circ + y^\circ + 70^\circ = 180^\circ $$
Since $( y = x^2 )$, we can substitute $( y )$ in the equation:
$$ x^\circ + x^2^\circ + 70^\circ = 180^\circ $$
Simplifying, we get:
$$ x^2^\circ + x^\circ = 110^\circ $$
This is a quadratic equation that we can solve for $( x )$:
$$ x^2^\circ + x^\circ - 110^\circ = 0 $$
This equation can be solved using the quadratic formula, but for simplicity, we can guess and check values of $( x )$ that are integers, because if $( y = x^2 )$, and $( x )$ and $( y )$ are angles, they must be less than 180 degrees. We can test some values:
If $( x = 10 )$:
$$ 10^2^\circ + 10^\circ = 100^\circ + 10^\circ = 110^\circ $$
This satisfies the equation. So, $( x = 10 )$ and $( y = x^2 = 10^2 = 100 )$.
Therefore, the value of $( y )$ is 100.
</think>
\boxed{100}

**GRPO**

<think>
We are given that the angle y is equal to the square of the angle x, and the angle x is given as 70 degrees. Therefore, we can set up the following equation:
y = x^2
Substituting 70 for x, we get:
y = 70^2
y = 4900
Therefore, the value of y is 4900.
</think>
\boxed{4900}

Figure 7: Sample generated responses by CPPO and GRPO. GRPO exhibits a perception error that is corrected in the CPPO response.

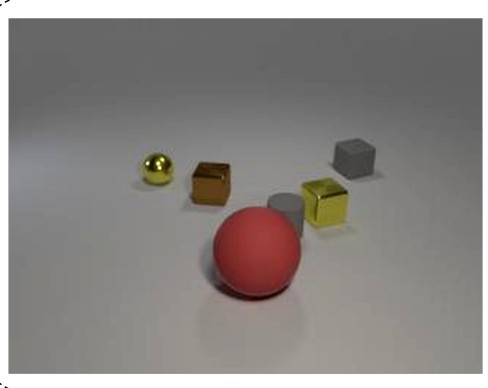

Hint: Please answer the question requiring an integer answer and provide the final value, e.g., 1, 2, 3, at the end.
Question: Subtract all brown cubes. Subtract all gray cylinders. How many objects are left?

**CPPO**

<think>
I have six objects: a yellow sphere, a brown cube, a red sphere, a gray cylinder, a yellow cube, and a gray cube. After removing the brown cube and gray cylinder, I am left with four objects.
</think>
\boxed{4}

**GRPO**

<think>
To answer the question, I'll follow these steps:
1. Identify the initial number of objects: 5 (yellow spherical object, brown cube, gray cube, gray cylinder, yellow cube)
2. Remove the brown cube: This leaves us with 4 objects
3. Remove the gray cylinder: This leaves us with 3 objects
Therefore, after removing both the brown cube and the gray cylinder, there are 3 objects left.
Final answer: 3
</think>
\boxed{3}

Figure 8: Sample generated responses by CPPO and GRPO. GRPO exhibits a perception error that is corrected in the CPPO response.

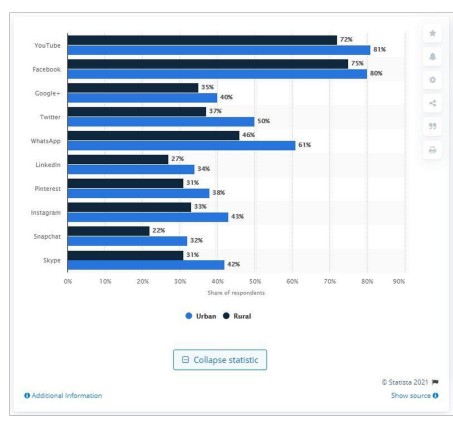

Hint: Please answer the question requiring an integer answer and provide the final value, e.g., 1, 2, 3, at the end.
Question: What is the difference between highest and lowest value of dark blue bar?

**CPPO**

<think>
To find the difference between the highest and lowest values of the dark blue bar, I need to identify these values from the chart.
From the chart:
- The highest dark blue bar represents Facebook, with a value of 75%.
- The lowest dark blue bar represents Snapchat, with a value of 22%.
The difference is calculated as:
75% - 22% = 53%
Therefore, the difference between the highest and lowest value of the dark blue bar is 53%.
</think>
\boxed{53}

**GRPO**

<think>
To determine the difference between the highest and lowest value of the dark blue bars:
1. Identify the highest value: YouTube has the highest dark blue value, which is 72.
2. Identify the lowest value: Snapchat has the lowest dark blue value, which is 22.
3. Calculate the difference: 72 - 22 = 50.
Therefore, the difference between the highest and lowest value of the dark blue bars is 50.
</think>
\boxed{50}

Figure 9: Sample generated responses by CPPO and GRPO. GRPO exhibits a perception error that is corrected in the CPPO response.

