# OpenReview forum: "CPPO: Contrastive Perception for Vision Language Policy Optimization"
_ICLR.cc/2026/Conference — ICLR 2026 Conference Withdrawn Submission_

### Official Review · Reviewer_nwec · 2025-10-31

**Soundness:** 3
**Presentation:** 3
**Contribution:** 3
**Rating:** 4
**Confidence:** 4

**Summary:**

The paper presents CPPO, a reinforcement learning fine-tuning method for VLMs that targets perception errors without separating perception and reasoning. CPPO detects perception tokens via entropy shifts when the input image is replaced by an information-removing perturbation. It applies a Contrastive Perception Loss (CPL) on these tokens, aligning token distributions between original and information-preserving views while contrasting with information-removing ones via an InfoNCE objective. CPL is combined with GRPO using advantage gating. Experiments on Qwen2.5-VL-3B/7B trained on ViRL39K show consistent improvements on benchmarks like LogicVista, MathVista, DynaMath, and MathVerse.

**Strengths:**

- The paper presents a well-defined method for locating perception tokens through entropy analysis and focuses optimization only on those regions, reducing unnecessary regularization on reasoning-related outputs.
- The auxiliary loss introduces only one weighting coefficient and applies advantage gating to stabilize training, showing good compatibility with existing RL pipelines.
- CPPO consistently improves over GRPO and other perception-aware baselines. Ablation studies confirm the benefits of restricting CPL to perception tokens, the choice of the Top-k threshold, and the use of advantage gating.

**Weaknesses:**

- While the perturbation design is central to CPPO’s effectiveness, the study only evaluates a specific configuration without examining how varying perturbation categories or intensity levels affect the learned perception robustness.
- CPL appears orthogonal to GRPO, raising the question of whether it can be used as a standalone objective. It would strengthen the paper if the authors could include an experiment evaluating CPL in isolation (without GRPO) to verify whether it independently contributes to perception-level improvement.
- The paper would benefit from a more detailed interpretation of how CPL conceptually and practically connects with GRPO, clarifying whether their integration yields genuine synergy or simply combines two orthogonal objectives.

**Questions:**

See above

---

### Official Review · Reviewer_jNmN · 2025-10-31

**Soundness:** 3
**Presentation:** 3
**Contribution:** 3
**Rating:** 4
**Confidence:** 5

**Summary:**

This paper introduces Contrastive Perception Policy Optimization (CPPO) to address the critical challenge of disentangling perception and reasoning errors in RL-based VLM fine-tuning. The method proposes an entropy-based technique to identify "perception tokens" and applies a targeted contrastive loss to enhance their robustness. While the problem is significant and the approach demonstrates modest empirical gains, the work suffers from three major weaknesses. First, its core token detection mechanism is not novel and inherits the unaddressed, fundamental flaws of prior perturbation-based techniques. Second, the entire CPPO apparatus appears to be an overly intricate and indirect solution to a problem created by the authors' foundational choice of a critic-less RL framework. Finally, the marginal performance improvements do not appear to justify the substantial increase in methodological complexity and the unverified assumptions upon which the method rests.

**Strengths:**

1.  **Significant Problem Formulation:** The paper clearly articulates a fundamental and important problem in multimodal learning. Differentiating between perception and reasoning failures is crucial for building more robust and interpretable VLMs, and the authors provide a clear motivation for why this is a limitation in current RL-based fine-tuning paradigms.

2.  **Novel Application of Contrastive Learning:** While the detection method may not be new, the application of a contrastive loss specifically on these dynamically identified tokens within an RL loop is a clever integration of different learning paradigms to tackle this specific problem.

3.  **Consistent Empirical Improvements:** The method demonstrates consistent, though small, performance gains over strong baselines across a variety of visual reasoning benchmarks. The qualitative examples effectively illustrate cases where CPPO corrects perception errors that plague the baseline model, providing intuitive support for the method's objective.

**Weaknesses:**

1.  **Inherited Flaws of a Non-Novel Detection Method:** The core idea of using input perturbations to identify salient model dependencies is a well-established principle in the model interpretability literature (e.g., similar concepts in methods, e.g., POVID [1], VCD [2], SeVA [3], and more recently, VPPO [4], [5]). The paper presents this as a novel contribution for VLM-RL, yet fails to acknowledge this prior art or, more importantly, address the known, fundamental issues with such approaches. Specifically, this entire class of methods has been shown to suffer from a lack of systematic evaluation on:
    *   **Human-Model Alignment:** There is no guarantee that the tokens the model identifies as "perceptually sensitive" via entropy shifts align with what a human would consider direct visual perception. The method may simply be capturing arbitrary model sensitivities.
    *   **Hackability:** The model is not incentivized to improve its true perception, but rather to become invariant to the *specific set of augmentations* used in the contrastive loss. It can learn to "game" the objective, creating a false sense of robustness without genuine improvement in visual grounding.
    *   **Perceptual Grounding vs. Sensitivity:** The detector identifies *sensitivity*, not necessarily correct *grounding*. A token's prediction may be highly sensitive to a visual feature that the model is consistently misinterpreting. CPPO would then reinforce this misinterpretation to be more robust.

2.  **A Complex Solution to a Self-Imposed Problem:** The entire CPPO apparatus appears to be a convoluted workaround for a limitation inherent in its chosen backbone RL algorithm, GRPO. The fundamental goal of RL is effective credit assignment. In VLM-RL, this includes assigning credit (or blame) to tokens based on how well they are grounded in the visual input.
    *   **The Role of the Critic:** This fine-grained, state-dependent credit assignment is precisely the designated role of a **critic** in an Actor-Critic framework. A powerful critic should learn a value function that implicitly understands whether the actor's generated tokens are perceptually sound.
    *   **The Void Created by GRPO:** By design, GRPO discards the critic for training stability, creating a methodological vacuum: it loses the primary mechanism for intelligent credit assignment.
    *   **CPPO as an Indirect Patch:** CPPO attempts to fill this vacuum with a complex, post-hoc apparatus. Instead of an integrated learning signal from a critic, it uses a separate, inference-based detection module and an auxiliary loss. This makes the learning signal indirect and potentially inefficient. In essence, CPPO acts like a coach who only analyzes a game tape after a win, whereas a perception-aware critic acts like a coach providing real-time feedback during every play, making learning far more direct and efficient.

3.  **Marginal Returns for High Complexity:** The reported performance gains, while consistent, are modest (e.g., an average absolute improvement of ~1.5% for the 7B model). When weighed against the substantial increase in complexity—a multi-stage process of perturbation, forward passes, entropy calculation, token tagging, and an auxiliary contrastive loss—the practical value is questionable. The lack of statistical significance testing (e.g., results over multiple seeds) further weakens the claim that these small margins represent a meaningful improvement.

[1] Aligning Modalities in Vision Large Language Models via Preference Fine-tuning

[2] Mitigating Object Hallucinations in Large Vision-Language Models through Visual Contrastive Decoding

[3] Self-Supervised Visual Preference Alignment

[4] SPOTLIGHT ON TOKEN PERCEPTION FOR MULTIMODAL REINFORCEMENT LEARNING

[5] On Epistemic Uncertainty of Visual Tokens for Object Hallucinations in Large Vision-Language Models

**Questions:**

1.  **Regarding the novelty and robustness of the detector:** How does your perception token detector compare to prior perturbation-based methods in the literature? More importantly, how do you ensure that the model is not simply "hacking" the augmentation scheme, and can you provide any evidence that the identified tokens align with human judgments of perceptual grounding?

2.  **Regarding the framework choice:** Could you provide a compelling justification for building this complex external module on top of a critic-less framework, rather than tackling the more fundamental challenge of creating a **perception-aware critic** within a standard Actor-Critic setup? A comparison to a strong AC baseline, where the critic's value function is conditioned on visual alignment, would be a crucial experiment.

3.  **Regarding failure analysis:** Can you provide an analysis of cases where CPPO fails? Specifically, are there instances where reinforcing perceptual "consistency" via the contrastive loss leads to a degradation in multi-step reasoning, or where the token detector is demonstrably wrong?

---

### Official Review · Reviewer_BTUV · 2025-10-31

**Soundness:** 3
**Presentation:** 2
**Contribution:** 3
**Rating:** 6
**Confidence:** 3

**Summary:**

This paper proposes CPPO, a reinforcement-learning finetuning method for VLMs that explicitly strengthens perception within end-to-end reasoning. The key idea is to (i) identify perception tokens by measuring each token’s entropy change when the input image is replaced by an information-removing perturbation, and (ii) apply a token-level contrastive loss (CPL) only on those perception tokens: the distribution under the original image is the anchor, an information-preserving view is the positive, and an information-removing view is the negative, implemented via an InfoNCE objective. Importantly, the contrastive loss is applied only to perception tokens from correct rollouts, and the method does not require extra CoT supervision or proprietary evaluators.
Experiments on multiple math/visual-reasoning benchmarks (LogicVista, MathVista, DynaMath, WeMath, MathVision, MathVerse, MMMU-Pro-Vision) using Qwen2.5-VL-3B/7B backbones show consistent gains over GRPO and recent perception-aware RL methods.

**Strengths:**

# Strengths:

1. Entropy-based selection of perception-dependent tokens plus an unsupervised token-level InfoNCE loss is a neat way to target visual grounding without architectural surgery or stepwise supervision. The paper clearly explains the anchor/positive/negative construction and why token-level contrast makes sense for perception.

2. CPPO’s CPL does not require additional CoT supervision or proprietary models, improving simplicity and scalability.

3. On Qwen2.5-VL-3B/7B, CPPO outperforms GRPO and recent methods (OpenVLThinker, Visionary-R1/SR1, PAPO, NoisyRollout, Look-Back) across diverse benchmarks.

**Weaknesses:**

# Weaknesses:

1. The entropy selection is heuristic. Using entropy increase as a proxy for perception dependence is intuitive but heuristic; some non-perception tokens could also exhibit entropy shifts. A deeper discussion of limitations/failure cases (show some demos) would strengthen the claim.

2. An algorithm box/flow diagram of the training loop would improve clarity. Overall, I feel the current version is a little bit hard to follow.

3. Since CPL targets perception tokens, analyze whether reasoning quality (non-perception tokens) is preserved or improved.

**Questions:**

# Questions

1. CPPO’s success hinges on how “information-preserving” and “information-removing” transforms are chosen. Could the authors provide more empirical discussion regarding the effects of the "preserving" perturbation?

---

### Note · Authors · 2025-11-14

I have read and agree with the venue's withdrawal policy on behalf of myself and my co-authors.